# Effects of Urbanization on Flowering Phenology, Pollination, and Reproductive Success in the Chiropterophilous Tropical Tree *Ceiba pentandra*

**DOI:** 10.3390/plants14111575

**Published:** 2025-05-22

**Authors:** Henry F. Dzul-Cauich, Miguel A. Munguía-Rosas

**Affiliations:** Laboratorio de Ecología Terrestre, Departamento de Ecología Humana, Centro de Investigación y de Estudios Avanzados del Instituto Politécnico Nacional (Cinvestav), Km. 6 Antigua Carretera a Progreso, Mérida 97310, Mexico; henry.dzul@cinvestav.mx

**Keywords:** bat pollination, *Ceiba pentandra*, flowering phenology, pollination, reproductive success, urbanization

## Abstract

Urbanization often negatively impacts pollinator abundance and richness; however, its effects on different pollination components and plant reproductive success are highly variable. Previous research efforts have also shown geographic and taxonomical bias, with non-insect-pollinated plant species in tropical cities underrepresented in the literature. Although bats represent the most persistent mammal group in urban ecosystems, studies addressing the effect of urbanization on chiropterophilous plants are scarce. Here, we addressed the impacts of urbanization on flowering phenology, pollination, and reproductive success in the chiropterophilous tree *Ceiba pentandra* (L.) Gaertn. (Malvaceae) in two major tropical cities of the Yucatan Peninsula. We found that urbanization has led to an earlier flowering phenology; however, no effect of urbanization was detected in the two pollination components evaluated: pollinator visitation rate and pollen deposition. Finally, the effects of urbanization on the reproductive success of *C. pentandra* were mixed. While marginally negative effects of urbanization were found in fruit set, positive effects were found in seed germination. These findings suggest that urban pollinators can provide similar levels of pollination services and thus lead to comparable reproductive success for *C. pentandra* in forests and cities.

## 1. Introduction

Urbanization is an extreme case of habitat transformation through which the original vegetation cover has been almost eliminated [1,2]. It is therefore one of the land-use changes with the greatest impact on biodiversity, species interactions, and ecosystem services [3,4]. Cities now host ca. 50% of the world’s human population, with an increasing trend evident [5]. A fact of concern is that the most rapid rates of urbanization occur in the world’s most biodiverse regions (i.e., tropical areas) of the global south [6,7,8]. Consequently, urbanization in the tropics has a disproportionate impact on local biodiversity [9,10].

Many studies have found that urbanization has detrimental effects on plant and animal populations and communities, in which the main drivers include habitat loss and fragmentation, the introduction of exotic species, heat stress, environmental pollution, and an elevated abundance of domestic predators [11,12,13]. Despite the generalized biodiversity loss due to urbanization, some species populations may persist in cities and provide important ecosystem services such as pollination [14,15,16]. Around 87.5% of seed plants rely on animals for sexual reproduction, a value that reaches 94% in the tropics [17]. Moreover, 75% of crop plant species rely to some extent on animal pollination [18]. Urban pollination has recently gained interest because this interaction is critical for the reproduction of plants in this novel ecosystem [16] and because urban outdoor agriculture can contribute to the food security of the local inhabitants, particularly in developing countries [19,20].

The prevailing environmental conditions in urban ecosystems are inhospitable for most pollinators, and the fast rates at which urban drivers occur could prevent their adaptive response [16,21]. Habitat loss may affect pollen flow and deposition via modifications in pollinator foraging behavior [12]. Moreover, some pollutants may impair the health and behavior of plants [22], as well as those of their pollinator partners [16]. Artificial light at night also affects the pollinators’ visual perception of flowers and acts to increase predation risk, which, as a result, may negatively impact the reproductive success of plants, particularly in those nocturnally pollinated [23,24]. The heat-island effect is a pervasive phenomenon in cities around the world [25]. In addition to constituting a source of stress for plants and pollinators [22], it often leads to the earlier blooming of plants, which could, in turn, desynchronize plant–pollinator interactions [25]. Pollinators that persist in cities often employ opportunistic strategies such as broad diets and can utilize exotic plants as food resources and/or use anthropogenic habitats as refuge and/or nesting sites [16,26]. The identification of such species and the resources they use, as well as their manner of use, is critical since it could allow for the effective design of pollinator-friendly cities [26,27].

The effect of urbanization on pollinators and pollination services has been the subject of recent reviews (e.g., [15,26,28,29]). However, most of these have focused on communities of insect pollinators, particularly bees (e.g., [15,28,30]). The multi-taxa reviews available suggest that urbanization has had an overall negative effect on different groups of pollinators (e.g., [14,26,29]). However, these reviews also shared a clear taxonomic bias, with bees overrepresented and bats poorly/unrepresented, as well as a geographic bias, since most studies were conducted in temperate cities (e.g., [14,26,29]). Therefore, more studies in tropical cities, where pollination systems are more specialized and non-insect pollinators have a greater contribution to plant reproduction, are therefore urgently required [29,31]. On the other hand, there is no clear trend regarding the effect of urbanization on the different components of pollination and the consequences on plant reproductive success [29]. This lack of a consistent effect could be the result of the many forms in which these variables are measured [4,29]. For example, in the most recent meta-analysis on the topic, no effect of urbanization was detected for the pollinator visitation rate or fruit set, while a negative effect was found for seed set [29]. The effects of urbanization on pollinator services should therefore be assessed in terms of several pollination components (e.g., visiting rate, pollen deposition) and different aspects of reproductive success (fruit and seed set), since these components could be affected differentially.

Bats pollinate more than 500 plant species in tropical and subtropical regions. Some of these species are dominant elements of plant communities or valuable crop plants [32,33]. The dependence of chiropterophilous plants on bats is high; it is estimated that fruit/seed set may drop 83% on average with the exclusion of bats [33]. Bats also represent the most diverse group of mammals remaining in urban ecosystems [13]. Phytophagous bats seem to thrive in urban ecosystems, mainly due to their facultative diet, which can include the fruit and nectar of native and exotic plants and, occasionally, insects [13,34]. Bats can thus provide valuable pollination and seed dispersal services (some bat species provide both services) in urban ecosystems [32,35]. However, accurate assessment of pollination and other services provided by bats in urban ecosystems is rare and is often simply inferred from the observation of bat visits [35].

Here, we addressed the effects of urbanization on the flowering phenology, pollination services, and plant reproductive success in a chiropterophilous tree, *Ceiba pentandra* (L.) Gaertn. (Malvaceae). To evaluate the spatial and temporal consistence of the effects, the study was conducted in two major tropical cities of the Yucatan Peninsula and over two reproductive seasons. In this region, *C. pentandra* is native and is commonly found in urban green areas. Given the complexity of the pollination process, we assessed the impact of urbanization on two different components of pollination (pollinator visitation rate and pollen deposition) as well as on different aspects of reproductive success (fruit and seed set as well as seed germination). Our specific research question was the following: What are the effects of urbanization on the flowering phenology, pollination, and reproductive success in *C. pentandra*? Given the variety of stressors found in the city that may affect the activity of pollinators, we predicted a decreased visiting rate of pollinators and a lower pollen load size, which could lead to reduced reproductive success. An earlier flower phenology in the city was also predicted, given the warmer temperatures commonly found in these ecosystems [25].

## 2. Results

### 2.1. Flowering Phenology

At the population level, apart from Merida in 2022 (Figure 1A), flowering started 3-18 days earlier in the city than in the forest (Figure 1B–D). Similarly, the population’s flowering peak consistently occurred earlier in the city than in the forest (Figure 1). The flowering season lasted longer in the city than in the forest in Merida and Cancun in 2022 (Figure 1A,B), but the opposite was observed in 2023 (Figure 1C,D). Similarly, in Merida and Cancun, the population’s flowering season finished 15 and 18 days earlier, respectively, in the forest than in the city in 2022 (Figure 1A,B). However, the opposite was true in 2023 (Figure 1C,D), in which the flowering season finished 27 and 30 days earlier in the city than in the forest in Merida and Cancun, respectively. The phenological overlap index (POI) between the city and forest populations was 0.95 in Merida and 0.88 in Cancun during the year 2022 but was 0.61 in Merida and 0.68 in Cancun during the year 2023.

At the individual level, flowering onset also occurred significantly earlier in the city (Julian day 32.16 ± 1.33 (mean values ± 1SE)) than in the forest (Julian day 45.45 ± 1.61), regardless of site or year (Table 1). Flowering onset also varied significantly between sites (Merida: 40.22 ± 1.16; Cancun: 36.23 ± 1.92) and years (2022: 32.72 ± 1.74; 2023: 43.43 ± 1.27); however, there was no significant interaction between these factors and urbanization (Table 1). Significant variation in the flowering duration was found between years (2022: 41.14 ± 1.74 days; 2023: 33.53 ± 1.27 days). Moreover, the interaction of urbanization × year was significant for this variable (Table 1); i.e., flowering of the trees lasted longer in the city than in the forest in 2022. However, the opposite trend (i.e., longer duration in the forest than in the city) was found in 2023 (Figure 2A). Termination of flowering occurred significantly later in the forest (Julian day 81.49 ± 1.46) than in the city (Julian day 70.45 ± 1.44) and was also significantly variable between sites (Merida: 79.15 ± 1.57; Cancun: 71.93 ± 1.48) and years (2022: 73.86 ± 1.64; 2023: 76.96 ± 1.49) (Table 1). Finally, the interaction of urbanization × year was significant in the year 2023 (Table 1), with differences in terms of flowering termination found between trees in the forest and city. However, this was not the case in the year 2022 (Figure 2B).

### 2.2. Floral Biology and Rewards

Regardless of habitat and site, the flowers opened early in the night (ca. 1900 h) and were found to contain some nectar at that time (Figure 3A,B). Pollen dehiscence occurred shortly (ca. 30 min) after anthesis, and nectar production also peaked immediately after anthesis (Figure 3A,B). No relevant differences were observed between city and forest or between sites in terms of the onset or peak of nectar production (Figure 3A,B). Both nectar volume (Figure 3A,B) and concentration (Figure 1C,D) were greater early in the night and steadily decreased as the night progressed. The lowest nectar volume (Figure 3A,B) and concentration (Figure 3 C,D) occurred during the morning of the following day (1000–1100 h). Lower nectar volume was observed in the forest than in the city during the morning in both sites (Figure 3A,B); however, these differences were more evident in Cancun than in Merida (Figure 3B). Nectar concentration was similar between the forest and the city most of the time; however, flowers in the city trees produced slightly more dilute nectar from 0300 to 0700 h in Merida and Cancun (Figure 3C,D). In both the forest and city, the flowers remained open, and the stigmas were apparently receptive (bright and turgent) by 1100–1200 h, with no significant difference between Merida and Cancun. Regardless of habitat or site, corolla dehiscence occurred between 1200 and 1300 h.

### 2.3. Pollination Components

While the pollinator visitation rate was unaffected by any of the sources of variation included in the model, pollen load size differed significantly between pollinator guilds and years (Table 2). However, although significant, the differences between these factor levels were small. Specifically, the pollen load size attributed to nocturnal visitors (362.66 ± 6.74 grains·stigma) was only 1.03 times greater than that attributed to diurnal visitors (351.06 ± 7.61 grains·stigma). Moreover, this same variable was only 1.02 times greater during the year 2022 (364.66 ± 6.86 grains·stigma) than during 2023 (372.68 ± 8.00 grains·stigma).

### 2.4. Reproductive Success

Fruit set was only marginally greater in the forest (24.65 ± 1.39%) than in the city (18.92 ± 1.36%) and was significantly greater (1.45 times) in nocturnally (25.71 ± 1.41%) than in diurnally (17.74 ± 1.30%) pollinated flowers (Table 2). The effect of the interaction of urbanization × pollinator guild on fruit set was also significant (Table 1); while diurnally pollinated flowers presented a significant reduction in fruit set in the city relative to that of the forest, this was not true of the nocturnally pollinated flowers (Figure 4A). Finally, fruit set was also 1.3 times greater in Cancun (25.87 ± 1.85%) than in Merida (19.84 ± 1.96%) and 1.82 times greater during the year 2022 (29.28 ± 1.83%) than during 2023 (16.03 ± 1.77%) (Table 2).

From all the sources of variation included in the model, only the effects of the year and the interaction of this variable with urbanization on the seed set were statistically significant (Table 1). Specifically, the seed set was 1.24 times greater during the year 2022 (88.46 ± 0.93%) than during 2023 (71.53 ± 2.21%) (Figure 4B). However, in the year 2022, the seed set was 1.07 times greater in the city than in the forest, but no difference between habitats was observed in 2023 (Figure 4B). Although not significant, the similarity in terms of seed set between diurnally (81.53 ± 1.49) and nocturnally (80.65 ± 1.49) pollinated flowers is noteworthy given the chiropterophilous syndrome of the flowers in the study species.

Significant effects of urbanization, pollinator guild, and year were found on seed germination (Table 2). Specifically, seed germination was 1.70 times greater in the city (48.75 ± 4.44%) than in the forest (28.75 ± 4.08%). Furthermore, the germination of seeds by nocturnally pollinated flowers (43.75 ± 1.36%) was 1.30 times greater than that of seeds sired by diurnally pollinated flowers (33.75 ± 2.59%). Finally, seed germination was 1.48 times greater in the year 2022 (46.25 ± 4.51%) than in 2023 (31.25 ± 4.14%) (Figure 5).

## 3. Discussion

This study investigated the effects of urbanization on the flowering phenology, two components of the pollination process, and the reproductive success of a chiropterophilous tree (*C. pentandra*) in two major tropical cities of the Yucatan Peninsula and the surrounding forest patches. According to the results, urbanization has led to an earlier flowering phenology and the production of slightly higher quantities of more diluted nectar. On the other hand, no significant effect of urbanization was detected on the components of pollination. Finally, the effects of urbanization on the reproductive success of *C. pentandra* in the Yucatan Peninsula were mixed; while marginally negative effects of urbanization were found in fruit set, positive and significant effects were found in seed germination. All these findings suggest that urban pollinators can maintain similar pollination services and reproductive success for *C. pentandra* in forests and cities. This is a positive finding for urban forests, since the maintenance of pollinator services for native plant species is highly important for maintaining the sexual reproduction and genetic diversity of plant populations in urban ecosystems.

Flowering phenology (at population and individual levels) exhibited an earlier start in the cities relative to the forest patches under study (see Table 1 and Figure 1). The same patterns have been identified in many plant species in several cities around the world (e.g., [25,36]). This trend is probably due to the warmer temperatures that are almost universally exhibited in urban areas, which are induced by the replacement of natural land cover with pavement, buildings, and other urban infrastructure (i.e., the urban heat-island effect [37]). This could certainly have been the case in our system since the temperature in the cities was ca. 4 °C warmer than in the forest (city: 30.84 ± 0.33 °C; forest: 26.66 ± 0.32 °C). However, given the observational nature of this research, we cannot rule out the possibility that other environmental variables that are concomitantly modified by urbanization could also have influenced this result (e.g., water availability and/or humidity [25,38]). A major concern when human-driven changes in flowering phenology occur is the desynchronization of plants with their pollinators, which may, in turn, negatively affect plant reproductive success [25]. Nevertheless, we did not detect significant negative consequences of the earlier flowering for pollination components or reproductive success in the urban populations of *C. pentandra* (Table 2). This could be because, although statistically different, the city and forest populations still exhibited an important overlap in their flowering phenologies at both the population (61 to 95%) and individual (66%) levels. This level of overlap, together with the high mobility of its major pollinator, the bat *A. jamaicencis* (whose daily movements range from 10 > 30 km from the roots [39,40]), suggests the occurrence of pollen flow between the city and forest. Although other descriptors of flowering phenology (i.e., flowering duration and termination) also exhibit some differences between the city and the forest, these effects were spatially or temporally inconsistent.

In terms of flowering biology, the most important difference observed was that the trees in the city produced slightly higher quantities of slightly more diluted nectar. Previous work has also found greater nectar production in urban relative to non-urban plant populations due to greater water availability in the urban ecosystems [41,42]. Flowers of trees growing under conditions of high levels of water availability may also have more diluted nectar since the nectar volume increases but not the amount of sugar. Water availability could therefore explain the greater production of more dilute nectar observed in the city in our system, since *C. pentandra* blooms during the dry season and is artificially irrigated in several urban green areas within the study area [43]. It should also be noted that differences in the volume and concentration of nectar were not only minor but these differences lasted for less than one-third of the full life span of the flowers. This is likely the reason why this difference is not associated with any important difference in any pollination component (Table 2).

Although urbanization is considered a major transformation of the habitat, we did not identify any effect of this process on the components of pollination evaluated (pollinator visitation rate and pollen load size). The major nocturnal pollinator of *C. pentandra* in the Yucatan Peninsula is *A. jamaicencis* (pollinator visitation rate: 1.08 ± 1.12 visits/h/inflorescence), a frugivore that may facultatively broaden its diet to include pollen, nectar, leaves, and even insects [44]. This bat species is also tolerant to a certain level of anthropogenic disturbance [45,46]. Similarly, Apidae bees (particularly the exotic *Apis mellifera*) are the major diurnal visitors of *C. pentandra* in the Yucatan Peninsula (visitation rate: 0.74 ± 1.05 visits/h/inflorescence) [47] and also have some tolerance to urbanization. For instance, *A. mellifera* is one of the most abundant bee species in other Neotropical cities [48]. This tolerance of the main nocturnal and diurnal pollinators to urbanization, together with the high density of *C. pentandra* seen in the cities of the Yucatan Peninsula [43], could explain why pollinator visitation rates and pollen deposition were unaffected by urbanization in our study. However, we recognize that further research is required in order to assess tolerance to urbanization and, as a first suitable approach, the assessment of pollinator abundance (nocturnal and diurnal) may be useful.

Although we found only a marginal negative effect of urbanization on fruit set, the significant interaction of urbanization × pollinator guild suggests that the reduction in fruit set attributable to diurnal visitors was the only factor significantly affected by urbanization (see Figure 4A). Since urbanization did not affect visitation rates or the pollen deposition of all visitors (Table 2), we can discard the possibility that the reduction in fruit set was due to negative effects of urbanization on these pollination components. One factor that could explain this result, but which remains to be explored, is that diurnal visitors move lower quantities of allogamous pollen in the city than in the forest. This is likely, since some degree of self-incompatibility is common in Neotropical populations of *C. pentandra* [49,50], and movement across the warmer inhospitable urban matrix could be less energetically profitable for bees than for bats, particularly when sufficient floral resources can be found in a single tree [49]. This likely leads to more geitonogamous crosses during the day in the flowers of urban trees and thus a lower fruit set, as seen in other plant species [51]. However, fruits of trees in the city exhibited similar seed set values to those of trees in the forest.

Interestingly, seeds from trees in the city presented significantly greater germination success than those from trees in the forest. We suggest that this was probably due to a reduction in the pre-dispersal seed predation of *C. pentandra* by hemipterans (*Dysdercus* sp.) in cities. This idea is supported by a previous study with *Ceiba aesculifolia*, in which the authors observed that forest disturbance reduced seed predation by *Dysdercus* sp. in another tropical dry forest of Mexico [52]. Although we observed considerably fewer *Dysdercus* sp. individuals predating seeds of *C. pentandra* in the city than in the forest, more systematic observations, along with a formal analysis, are required to test this hypothesis. However, regardless of the mechanism underlying the differences, the fact that negative and positive effects on different pollination components were found suggests that the net effect of urbanization on reproductive success could be neutral.

In conclusion, urbanization has led to earlier flowering. However, despite this difference, the pollinator visitation rate and pollen load size has been unaffected by urbanization. While marginally negative effects of urbanization were identified in fruit set values attributable to diurnal visitors, a positive effect of the same process was seen in seed germination. This is a positive result, since it suggests that urban pollinators in the study area can maintain the pollination services and sexual reproduction of the study species, which is native and of considerable cultural and ecological value to the region.

## 4. Materials and Methods

### 4.1. Study System

The study species (*Ceiba pentandra*) is an emergent tree with a pantropical distribution [53]. Flowering occurs at the beginning of the year [54] in the form of a brief and massive bloom [49]. The flowers exhibit nocturnal anthesis and are typically pollinated by bats (e.g., [55,56]), although diurnal pollinators such as bees also contribute to its reproductive success in the Yucatan Peninsula [47]. *Ceiba pentandra* produces large quantities of nectar (up to 10 L per tree), and the seeds are wind-dispersed [57]. In the Yucatan Peninsula, this tree species occurs in tropical dry forest areas with variable degrees of disturbance, as well as in urban green areas, where it is a dominant tree species and is either planted or naturally recruited [43].

The study area comprised the two main cities (located 300 km apart) of the Yucatan Peninsula: Merida (21° 2′3.68″ N, 89° 45′52.21″ W; 10–14 m a.s.l) and Cancun (21°1′47.43″ N, 87° 7′3.42″ W; 8–10 m a.s.l.), as well as the adjacent forested areas. The most characteristic tree species in the forested areas were *Lysiloma latisiliquum* (Fabaceae), *Piscidia piscipula* (Fabaceae), and *Bursera simaruba* (Burseraceae) [56]. Similarly, in the urban area green areas, *P. piscipula*, *B. simaruba*, and *Leucaena leucocephala* (Fabaceae) were the dominant native tree species [43]. The weather in the study area is tropical subhumid with summer rains [58]. The distances between habitats (forest to city) were 21 km in Merida and 26 km in Cancun. In both sites, a random sample of 100 reproductive trees (50 per habitat (forest and city)) with accessible branches (up to 5 m in height from ground level) and at least 2 km apart was selected, yielding a total sample size of 200 trees. The study lasted for two consecutive flowering seasons, in the years 2022–2023 and 2023–2024. Hereafter, for simplicity, these two seasons are referred to only as 2022 and 2023, respectively.

### 4.2. Flowering Phenology

To describe floral phenology at the population and individual levels in the two habitats (city and the forest) of both sites (Merida and Cancun) over two reproductive seasons (2022 and 2023), for each selected tree, the whole crown was visually scanned using binoculars for the presence or absence of flowering buds and open flowers. These observations were conducted twice a week from the first day of December until the end of the flowering season. This sampling effort was the most cost-effective choice because while flowering phenology is relatively short (4–5 weeks), increasing the frequency of monitoring dramatically increases the cost.

The floral phenology pattern at the population level in each site and year was analyzed graphically, and the temporal overlap of the phenological curves was then compared between forest and city using a phenological overlap index (POI) based on the Morisita–Horn’s dissimilarity index and relative to the proportion of trees during the whole reproductive season [59]. The POI ranges from 0 to 1, with one denoting maximum overlap and zero signifying no overlap at all [59]. At the individual level, three phenological descriptors per individual were calculated: flowering onset, termination, and duration. Flowering onset and termination were expressed in Julian days, with December 1st as day number one. Flowering duration was expressed as the number of days in which a tree produced flowers. The effects of urbanization (forest vs. city), site (Merida vs. Cancun), and year (2022 vs. 2023) on the three phenological descriptors were assessed with linear mixed-effects models (three models in total; one per descriptor). In each model, the plant was included in the random part of the model to account for repeated measures. Only second-order interactions involving urbanization were included in the models. Our interest in the interactions was to assess if the effects of urbanization on phenology were spatially and/or temporally consistent.

### 4.3. Floral Biology and Rewards

A group of 40 flowers from two different trees per habitat and site (n = 40 flowers × 2 trees × 2 habitats × 2 sites = 320 flowers) were labeled, isolated with a mesh bag (hole size: 2 mm), and observed every two hours from the bud stage until flower wilting. For each flower, the time to anthesis, pollen dehiscence, stigma receptivity, and corolla abscission were recorded. Stigma receptivity was determined by appearance (turgor, color, etc.) as well as the stigma response after adding two drops of water [60]. For the same number, but a different group, of flowers from the same inflorescence and tree, the nectar volume and concentration were measured every hour from anthesis until the nectar production stopped. The nectar volume was measured with a hypodermic syringe and the nectar concentration was measured with a portable hand refractometer (Hanna instruments, HI96832, Washington, DC, USA). For logistical reasons, floral biology and rewards were only measured during the year 2023.

### 4.4. Pollinator Components

Two components of pollination were measured in the study species (pollinator visitation rate and pollen load size) in the two habitats, sites, and years. To assess the pollination visitation rate, we used the approach outlined by Dzul-Cauich and Munguía-Rosas [23] in the same tree species and site. In short, nocturnal visitors in one inflorescence per tree with 10–34 flowers were video-recorded with a Sony Handy Cam, using the night shot function (DCR-SR85, Beijing, China) and an independent infrared lamp (IRLamp6, Wildlife Engineering, Change to La Crosse, WI, USA) as a supplementary source of light. Video recordings began with anthesis and lasted for two hours. Since we were aware that diurnal visitors are effective pollinators of *C. pentandra* in the study area, the visitation rate of these visitors was also assessed using the same camera but with natural light. Diurnal filming began at sunrise (ca. 0600 h) and lasted for the same period as for nocturnal visitors. The videos were examined in slow motion and the number of visits as well as the feeding behavior was recorded. Visitors that contacted the reproductive organs of the flowers were considered effective pollinators. The same trees filmed during the night were filmed the following morning. In total, 272 and 288 inflorescences and trees were filmed in the forest and city, respectively.

Since two contrasting guilds of pollinators (diurnal vs. nocturnal) are involved in the sexual reproduction of *C. pentandra* in the Yucatan Peninsula, the two components of pollination were also assessed for diurnal and nocturnal pollinators separately. For this purpose, two exclusion treatments were conducted: (i) Nocturnal pollination: flowers of accessible inflorescences were exposed only to nocturnal visitors and covered with mesh bags (hole size: 2 mm) from sunrise (ca. 0600 h) until dehiscence of the corollas. (ii) Diurnal pollination: flowers were covered at the mature bud stage with mesh bags all night and uncovered the following morning (from 0600 h until corolla dehiscence) to be exposed only to diurnal visitors. Three flowers per inflorescence per treatment combination (n = 3 flowers × 50 trees × 2 sites × 2 habitats × 2 years = 1200 flowers) were collected and fixed in a solution of formaldehyde: acetic acid: ethanol and transported to the laboratory. The flowers were dissected to extract the styles and then softened in 1 N KOH at 65 °C for 20 min, rinsed with distilled water, and stained for 20 min at 65 °C in aniline blue [60]. Finally, the stained styles were mounted individually on glass slides. For each style, the number of conspecific pollen grains on the stigma was quantified under a fluorescence microscope (Leica DM2500LED, Wetzlar, Germany).

The effects of urbanization (forest vs. city), pollinator guild (diurnal vs. nocturnal pollinators), site (Merida vs. Cancun), and year (2022 vs. 2023) on the pollinator visitation rate and pollen load size (two separate models) were assessed with generalized mixed-effects models with Poisson error distribution. To determine whether the effect of urbanization found for these two variables was contingent upon the pollinator guild, site, or year, second interactions involving urbanization were included in both models. In each model, the plant was included in the random part of the model to account for spatial and temporal correlation.

### 4.5. Reproductive Success

To measure reproductive success due to the main pollinator guilds (diurnal vs. nocturnal) in the two habitats, sites, and years, a random sample of seven flowers per inflorescence was selected and allocated to each of the two pollinator exclusion treatments described in Section 2.4 (n = 7 flowers × 193 trees × 2 habitats × 2 sites × 2 years = 1351 flowers per treatment). These tagged flowers were checked once a week until the flower either aborted or set fruit. The proportion of fruit produced in the sample was then used to calculate the fruit set. To assess the seed set, a group of 5–10 mature fruits per sampled tree was collected and opened to count the seeds. Since the mean number of ovules in the study population is known (314 ovules [23]), the seed set value was obtained by dividing the number of seeds by the mean number of ovules. The total sample size was 578 fruits (321 in the forest and 257 in the city).

From the fruits collected, 50–100 apparently viable seeds per fruit were taken and the seeds from each treatment combination were then pooled. A sample of 50 seeds from each pool was randomly selected. These seeds were scarified with fine sandpaper (180–200 grit) and disinfected with a 1.5% solution of hypochlorite. The seeds were also sprayed with a fungicide (quaternary of ammonium compounds and glutaraldehyde; ANIBAC Cítrico; Promotora Técnico Industrial, Jiutepec, Mexico). Groups of five seeds from each treatment combination were placed in Petri dishes (n = 800 seeds in 160 dishes, 10 true replicates per treatment combination) with a disc of moist filter paper at the bottom. The seeds were kept in a growth chamber (Binder Inc., KBWF 720, Tuttlingen, Germany) at a constant temperature of 26 °C and a photoperiod of 12 h light and 12 h dark. Light was provided by high-pressure sodium lamps (PAR = 56.59 μmol·m^2^·s^−1^). Seed germination (radicle emergence) was then recorded every 48 h for 30 days.

The effects of urbanization, pollinator guild, site, and year, as well as the second interaction involving the effects of urbanization on fruit set, seed set, and germination (three models in total), were assessed using generalized mixed-effect models with binomial error distribution. As with the previous models, the plant was included in the random part of each model to account for spatial and temporal correlations. In the case of germination, accumulated seed germination after 30 days was used as the response variable.

All analyses were conducted in R 4.0.3 [61]. The raw data are presented in Appendix A.

## Figures and Tables

**Figure 1 plants-14-01575-f001:**
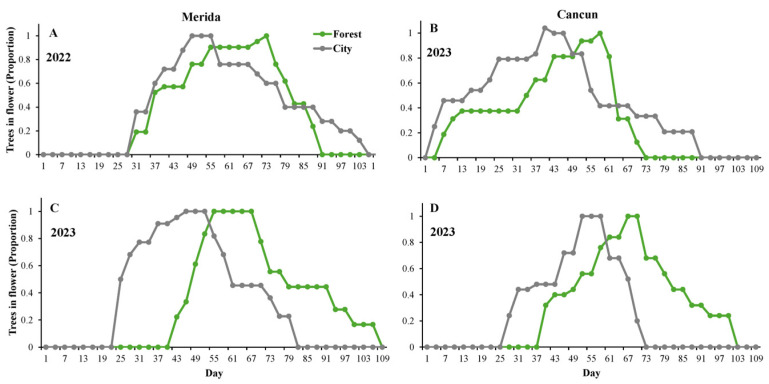
Flowering phenology of *Ceiba pentandra* in two localities in the Yucatan Peninsula (Merida (**A**,**C**) and Cancun (**B**,**D**)) recorded over two consecutive flowering seasons (2022 (**A**,**B**) and 2023 (**C**,**D**)). The lines show the number of flowering trees in two habitats: forest (in green) and city (gray). The horizontal axis shows the Julian day, in which December 1st = day 1.

**Figure 2 plants-14-01575-f002:**
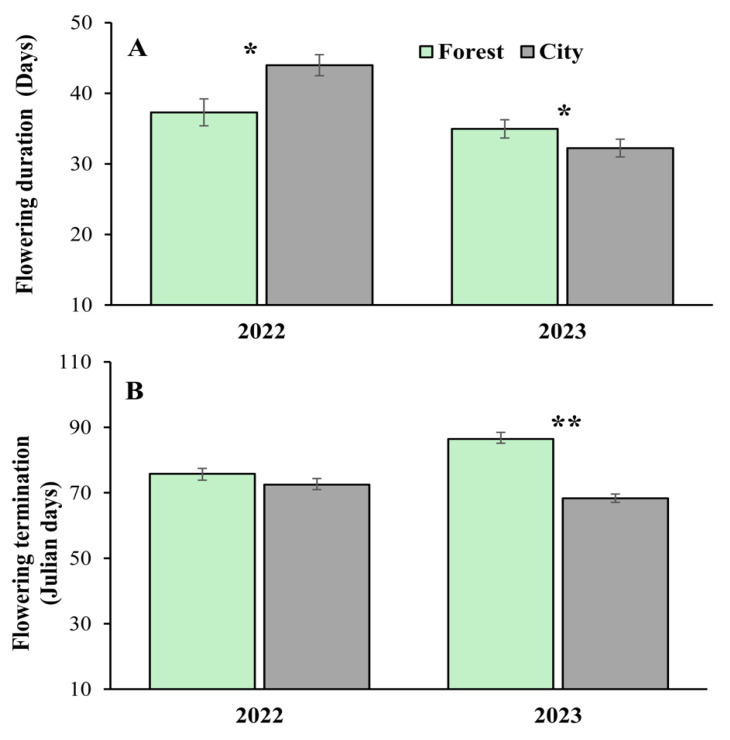
Flowering duration (**A**) and date of flowering termination (**B**) in *Ceiba pentandra* in two contrasting habitats (forest and city) over two consecutive reproductive seasons (years 2022 and 2023). The bars denote mean values ± 1SE. The asterisks indicate that the means differ statistically (* *p* < 0.05; ** *p* < 0.01).

**Figure 3 plants-14-01575-f003:**
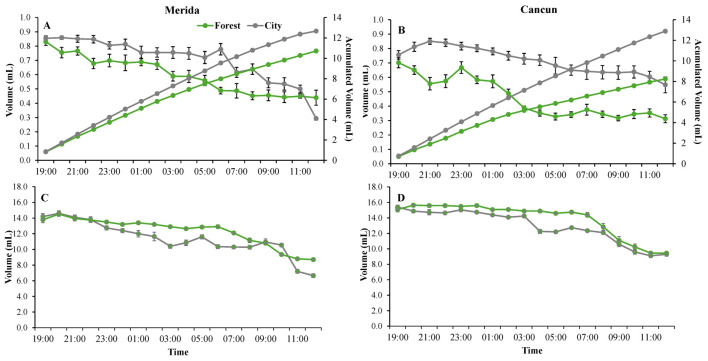
Nectar production in flowers of *Ceiba pentandra* in two contrasting habitats (forest and city) in two sites in the Yucatan Peninsula: Merida (**A**) and Cancun (**B**). Values of nectar volume, measured every hour, are shown along the main axis, while the accumulated volume is shown along the secondary axis. Values of nectar concentration in the forest and city in Merida (**C**) and Cancun (**D**) are also shown. All data are mean values. When shown, error bars denote ± 1SE.

**Figure 4 plants-14-01575-f004:**
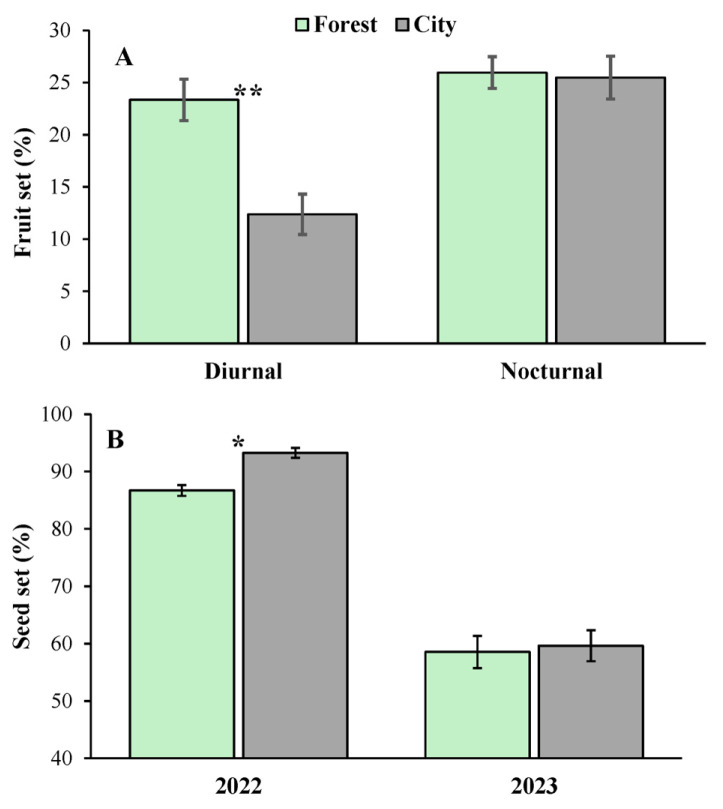
(**A**) Fruit set of *Ceiba pentandra* attributed to diurnal (Diurnal) and nocturnal pollinators (Nocturnal) in two contrasting habitats (forest and city) in the Yucatan Peninsula. (**B**) Seed set of *Ceiba pentandra* in two contrasting habitats (forest and city) over two consecutive reproductive seasons (years 2022 and 2023). The bars show mean values ± 1SE. The asterisks indicate statistically different pairs of means (* *p* < 0.05; ** *p* < 0.01).

**Figure 5 plants-14-01575-f005:**
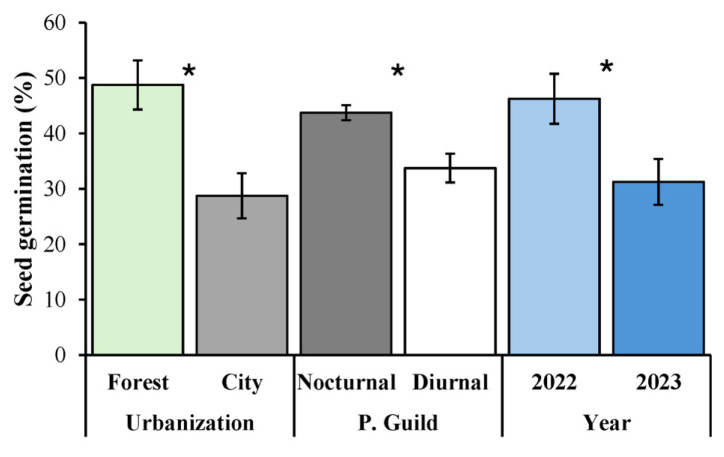
Germination of *Ceiba pentandra* seeds from different habitats (forest vs. city) pollinated by different pollinator guilds (nocturnal vs. diurnal) and over two different years (2022 and 2023). The bars show mean values ± 1SE. The asterisks indicate statistically different pairs of means (* *p* < 0.05) for each factor: urbanization, pollinator guild (*P. guild*), and year.

**Table 1 plants-14-01575-t001:** Results of statistical analyses to determine the effects of urbanization (forest vs. city), site (Merida vs. Cancun), and reproductive season (year 2022 vs. 2023) on three descriptors of the *Ceiba pentandra* flowering phenology at the individual level in the Yucatan Peninsula: flowering onset (Onset), flowering duration (Duration), and flowering termination (Termination). All data are the result of a Wald chi-square test with one degree of freedom.

		Response	
Source of Variation	Onset	Duration	Termination
Urbanization	χ^2^_1_ = 157.19 **	χ^2^_1_ = 1.59	χ^2^_1_ = 122.56 **
Site	χ^2^_1_ = 24.38 **	χ^2^_1_ = 3.84	χ^2^_1_ = 53.38 **
Year	χ^2^_1_ = 69.29 **	χ^2^_1_ = 24.27 **	χ^2^_1_ = 5.02 *
Urbanization × Site	χ^2^_1_ = 0.04	χ^2^_1_ = 0.01	χ^2^_1_ = 0.12
Urbanization × Year	χ^2^_1_ = 1.56	χ^2^_1_ = 10.75 **	χ^2^_1_ = 25.92 **

* *p* < 0.05 ** *p* < 0.01.

**Table 2 plants-14-01575-t002:** Results of statistical analyses to determine the effects of urbanization (forest vs. city), pollinator guild (*P. guild*: diurnal vs. nocturnal), site (Merida vs. Cancun), and reproductive season (2022 vs. 2023) on two components of pollination (pollinator visitation rate (Visits) and pollen load size (Pollen load)) and reproductive success (fruit and seed set, as well as seed germination) of *Ceiba pentandra* in the Yucatan Peninsula. All data are the result of a Wald chi-square test with one degree of freedom.

	Response				
Source of Variation	Visits	Pollen Load	Fruit Set	Seed Set	Germination
Urbanization	χ^2^_1_ = 0.06	χ^2^_1_ = 1.19	χ^2^_1_ = 3.83 †	χ^2^_1_ = 2.69	χ^2^_1_ = 4.76 *
*P. guild*	χ^2^_1_ = 0.46	χ^2^_1_ = 37.19 **	χ^2^_1_ = 16.22 **	χ^2^_1_ = 0.46	χ^2^_1_ = 4.23 *
Site	χ^2^_1_ = 0.01	χ^2^_1_ = 0.79	χ^2^_1_ = 8.95 **	χ^2^_1_ = 2.14	χ^2^_1_ = 3.12
Year	χ^2^_1_ = 0.06	χ^2^_1_ = 11.14 **	χ^2^_1_ = 34.79 **	χ^2^_1_ = 66.76 **	χ^2^_1_ = 5.46 *
Urbanization × *P. guild*	χ^2^_1_ = 0.95	χ^2^_1_ = 2.36	χ^2^_1_ = 16.30 **	χ^2^_1_ = 0.04	χ^2^_1_ = 0.01
Urbanization × Site	χ^2^_1_ = 0.11	χ^2^_1_ = 0.61	χ^2^_1_ = 0.19	χ^2^_1_ = 0.01	χ^2^_1_ = 0.05
Urbanization × Year	χ^2^_1_ = 1.06	χ^2^_1_ = 2.02	χ^2^_1_ = 0.14	χ^2^_1_ = 15.59 **	χ^2^_1_ = 0.07

† *p* = 0.05; * *p* < 0.05; ** *p* < 0.01.

## Data Availability

All data used in this study are available as Appendix A.

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
