# Peer review of "Effects of Urbanization on Flowering Phenology, Pollination, and Reproductive Success in the Chiropterophilous Tropical Tree Ceiba pentandra"

_plants, 2025, doi:10.3390/plants14111575_

Round 1
Reviewer 1 Report
Comments and Suggestions for Authors
This is an interesting and topical paper, within the scope of Plants. It relates to habitat transformation, urbanization in particular, and its effects on flower traits, and pollinator foraging behavior, and plant reproductive success a lot of work. Regarding the manuscript, I mention below some points to consider:
Abstract
Line 23. “Thus, these positive effects may offset the negative impact on reproductive success”. Please reorganize this sentence. This conclusion cannot be deduced from the current data.
Results
Figures, the resolution of the figures is insufficient. It should exceed 300 dpi.
Present the visitation frequencies of bats and honey bees, respectively.
Discussion
The abundance of pollinators directly affects the reproductive success of plants. The author must elucidate whether the abundance of pollinators in urban environments and woodlands is comparable.
Materials and Methods
Lines 365-366. Why did the authors conduct twice a week?
Lines 391-392. How to rule out the effects of pollinators? The bats may collect the nectar before the measurements.
Lines 417-419. Mesh bags! What is the hole size of the mesh bags?
Author Response
|
Comment 1: This is an interesting and topical paper, within the scope of Plants. It relates to habitat transformation, urbanization in particular, and its effects on flower traits, and pollinator foraging behavior, and plant reproductive success a lot of work. Regarding the manuscript, I mention below some points to consider: Abstract. Line 23. “Thus, these positive effects may offset the negative impact on reproductive success”. Please reorganize this sentence. This conclusion cannot be deduced from the current data. |
|
Response 1: We agree with the reviewer regarding that the assertion that the line “positive effects [of urbanization] may offset the negative impact on reproductive success” may need additional support/analysis. Therefore, in this new version of the manuscript, we now state only that “…the effects of urbanization on the reproductive success of C. pentandra were mixed” and deleted the sentence pointed out by the Reviewer. Please see Lines 20-23, 254-258, and 343-345. |
|
Comment 2: Results. Figures, the resolution of the figures is insufficient. It should exceed 300 dpi. |
|
Response 2: As the reviewer suggested, we are providing high-resolution figures (Tiff format & 300-400 dpi) as separate files with this new version of the manuscript. Comment 3: Present the visitation frequencies of bats and honey bees, respectively. Response 3: The information requested by the reviewers has been added. Please Lines 302-308. Comment 4: Discussion. The abundance of pollinators directly affects the reproductive success of plants. The author must elucidate whether the abundance of pollinators in urban environments and woodlands is comparable. Response 4: The main focus of our study was pollination and the reproductive biology of C. pentandra (Lines 96-98), not the pollinator community. In other words, our approach was phytocentric. We recognize that, as mentioned by the reviewer, pollinator abundance may provide some additional relevant information such as pollinator’s tolerance/susceptibility to urbanization; however, we did not record this variable. We have explicitly acknowledged this limitation of our study. Please see lines 310-315. Comment 4: Lines 365-366. Why did the authors conduct twice a week? Response 4: This was the most cost/effective choice. While flowering phenology is relatively short (4-5 weeks), the frequency of monitoring dramatically increases the cost (200 trees in two different locations and for two years were considered in this study). See lines 380-383. Comment 5: Lines 391-392. How to rule out the effects of pollinators? The bats may collect the nectar before the measurements. Response 5: We involuntarily omitted this detail. In this version of the manuscript, we have stated that the flowers were covered with a mesh bag (hole size 2 mm) prior to nectar measurements (Lines 402-404). Comment 5: Lines 417-419. Mesh bags! What is the hole size of the mesh bags? Response 5: The diameter of the holes of the mesh was ca. 2 mm. Please see lines 403-404 & 437.
|

Reviewer 2 Report
Comments and Suggestions for Authors
My comments on the manuscript “Effects of urbanization on flowering phenology, pollination, and reproductive success in the chiropterophilous tropical tree Ceiba pentandra”.
1. The rationale of choosing two years for data collection was not established, and why only two years?
2. The climatic factor characteristics (temperature, rainfall amount and regime, PET, dry season length, etc.) of each of the two years were not presented as explanatory factors of the phenological variation trends noticed for the two consecutive years of experimentation, and that matters.
3. The authors indicated that in cities some Ceiba pentandra trees were artificially irrigated in the dry season, but that factor was not taken into account in the sampling design. That is a bias so the authors should remove these irrigated trees from the samples.
4. At Line 294 the correct scientific name is Apis mellifera not Apis mellefera.
5. At Line 291 Write the full name of Artibeus jamaicensis not Artibeus jamaiciensis.
Author Response
Comment 1: My comments on the manuscript “Effects of urbanization on flowering phenology, pollination, and reproductive success in the chiropterophilous tropical tree Ceiba pentandra”.
The rationale of choosing two years for data collection was not established, and why only two years?
Response 1. As requested by the reviewer, we have added the rationale behind the assessment of two reproductive seasons (and sites). The year was not relevant as a main source of variation in our study; instead, the interaction of this variable with urbanization was more interesting for us. In other words, we included the year factor to assess how generalizable, or temporally consistent, can be the effects of urbanization. Please see lines 96-100 and 380-383.
On the other hand, since the interaction urbanization x year was not significant for the majority of our variables (See Table 2); we did not consider it cost-effective to extend the duration of the study.
Comment 2. The climatic factor characteristics (temperature, rainfall amount and regime, PET, dry season length, etc.) of each of the two years were not presented as explanatory factors of the phenological variation trends noticed for the two consecutive years of experimentation, and that matters.
Response 2. While we agree with the reviewer that addressing the influence of environmental variables on interannual variation on the flower phenology is an interesting topic, this is beyond the scope of our study. Our main objective was to assess the effect of urbanization on phenology. Temporal variation of phenology per se, was not relevant for us. However, as suggested by the reviewer, we invoked air temperature as a potential explanatory factor of earlier flowering in the cities that in the forests (city: 30.84±0.33 °C, forest: 26.66±0.32 °C) (Lines 269-271). We obtained the data from the nearest weather stations; however, we do not have data at the tree level, so we were unable to include this as proper explanatory continuous variable in the statistical analyses. Temperature has long been invoked as the cause of earlier flowering phenology in cities (see references 25, 36 and 37), although other environmental variables, such as those mentioned by the reviewer, can be also involved, these are usually correlated with air temperature and thus, its effect cannot be controlled in observational studies like ours (Lines 270-273).
Comment 3. The authors indicated that in cities some Ceiba pentandra trees were artificially irrigated in the dry season, but that factor was not taken into account in the sampling design. That is a bias so the authors should remove these irrigated trees from the samples.
Response 3. Nearly all sampled trees in urban habitats were influenced, to some extent, by artificial irrigation. Artificial irrigation could not be avoided during the study because it is an essential part of the management of these areas (particularly for ornamental plants). Moreover, removing irrigated trees from our data set may dramatically reduce our sample size. Nevertheless, we wish to mention that artificial irrigation of urban green areas is one of the most common practices around the globe; in this sense, artificial irrigation is not a source of bias since it could be considered as inherent to urbanization (see for example Marchianoni et al. 2021. Landscape and Urban Planning, 2015: 104198).
Comment 4. At Line 294 the correct scientific name is Apis mellifera not Apis mellefera.
Response 4. We have amended this error highlighted by the reviewer (see Lines 306-307).
Comment 5. At Line 291 write the full name of Artibeus jamaicensis not Artibeus jamaiciensis.
Response 5. We have amended this error highlighted by the reviewer (see Lines 281-282, 303).

Reviewer 3 Report
Comments and Suggestions for Authors
The authors investigated the effects of urbanization on flowering phenology, pollination, and pollination success of a chiropterophilous tree, Ceiba pentandra, in two major tropical cities and surrounding forest patches. They found that marginally negative effects of urbanization were found in fruit sets, but positive and significant effects were found in seed germination.
The Abstract and Introduction are well-written, but I have several concerns regarding the Results. First, I suggest to present the results of seed germination in Figure 4. Secondly, please provide data about pollinator assemblages in the daytime, given they are as important as bats for seed sets.
I also have some minor comments.
L99 and L235-236, I don’t think pollen load or pollen deposition is a qualitative component of pollination.
L105-106, put the sentence before your specific research question
L131, “32.16±1.33 day” , “45.45±1.61 day”
L143, x should be × instead here and in your tables.
L173-174, lack statistic analysis result support
There is no significance level in most of your figures
Author Response
Comment 1. The authors investigated the effects of urbanization on flowering phenology, pollination, and pollination success of a chiropterophilous tree, Ceiba pentandra, in two major tropical cities and surrounding forest patches. They found that marginally negative effects of urbanization were found in fruit sets, but positive and significant effects were found in seed germination. The Abstract and Introduction are well-written, but I have several concerns regarding the Results. First, I suggest to present the results of seed germination in Figure 4.
Response 1. As requested, the significant results of seed germination are now shown in a figure. However, Figure 4 was designed to show the interaction between two variables. Since no interaction was significant for seed germination, we decided to show these data as a new figure. Please see Figure 5.
Comment 2. Secondly, please provide data about pollinator assemblages in the daytime, given they are as important as bats for seed sets.
Response 2. As the reviewer requested, we have included data regarding the seed set values attributable to diurnally and nocturnally pollinated flowers (see Lines 225-228).
Comment 3. L99 and L235-236, I don’t think pollen load or pollen deposition is a qualitative component of pollination.
Response 3. To attend the concern of the reviewer, we have eliminated the terms qualitative and quantitative components of pollination from the text. Instead, pollinator visitation rate and pollen load size are now referred only as “two pollination components” throughout the manuscript. See for example Lines 10-11, 101-102, 253-255 179-181, and elsewhere.
Comment 4. L105-106, put the sentence before your specific research question
Response 4. We are grateful for the suggestion of the reviewer; however, we would like to keep our prediction (the sentence previously in L 105-106) after the research question. This is because the prediction provides a potential answer to our question. From our perspective, it may be clearer for the reader if the prediction follows the research question.
Comment 5. L131, “32.16±1.33 day” , “45.45±1.61 day”
Response 5. These data refer to Julian days, not to the days as units. We have added the text “Julian Day…” to avoid any misinterpretation (Lines 128-130).
Comment 6. L143, x should be × instead here and in your tables.
Response 6. We have placed the correct symbol throughout the text where appropriate, as suggested by the reviewer. Please see lines 141, 317, 402-404, 441 and Tables 1 & 2.
Comment 7. L173-174, lack statistic analysis result support.
Response 7. Descriptions of floral biology are commonly only qualitatively assessed, particularly in canopy trees where nectar measurement is challenging. See for example references 47 & 49 in the manuscript.
Comment 8. There is no significance level in most of your figures.
Response 8. As suggested by the reviewer, we have added the alpha level to the figures, where appropriate. See Figures 2,4 & 5 and respective captions.
